# Lytic Spectra of Tailed Bacteriophages: A Systematic Review and Meta-Analysis

**DOI:** 10.3390/v16121879

**Published:** 2024-12-04

**Authors:** Ivan M. Pchelin, Andrei V. Smolensky, Daniil V. Azarov, Artemiy E. Goncharov

**Affiliations:** 1Department of Molecular Microbiology, Institute of Experimental Medicine, Saint Petersburg 197022, Russia; denazarov.da@gmail.com (D.V.A.); phage1@yandex.ru (A.E.G.); 2Department of Computer Science, Neapolis University Pafos, Paphos 8042, Cyprus; andrei.smolensky@gmail.com

**Keywords:** review, meta-analysis, bacteriophage, intraspecies host range, *Caudoviricetes*, fitness cost

## Abstract

As natural predators of bacteria, tailed bacteriophages can be used in biocontrol applications, including antimicrobial therapy. Also, phage lysis is a detrimental factor in technological processes based on bacterial growth and metabolism. The spectrum of bacteria bacteriophages interact with is known as the host range. Phage science produced a vast amount of host range data. However, there has been no attempt to analyse these data from the viewpoint of modern phage and bacterial taxonomy. Here, we performed a meta-analysis of spotting and plaquing host range data obtained on strains of production host species. The main metric of our study was the host range value calculated as a ratio of lysed strains to the number of tested bacterial strains. We found no boundary between narrow and broad host ranges in tailed phages taken as a whole. Family-level groups of strictly lytic bacteriophages had significantly different median plaquing host range values in the range from 0.18 (*Drexlerviridae*) to 0.70 (*Herelleviridae*). In *Escherichia coli* phages, broad host ranges were associated with decreased efficiency of plating. Bacteriophage morphology, genome size, and the number of tRNA-coding genes in phage genomes did not correlate with host range values. From the perspective of bacterial species, median plaquing host ranges varied from 0.04 in bacteriophages infecting *Acinetobacter baumannii* to 0.73 in *Staphylococcus aureus* phages. Taken together, our results imply that taxonomy of bacteriophages and their bacterial hosts can be predictive of intraspecies host ranges.

## 1. Introduction

Tailed bacteriophages are a monophyletic group of viruses and belong to the class *Caudoviricetes* [1]. Accounting for a predominant part of studied bacteriophage diversity [2,3], they accompany their bacterial hosts in all habitats, including seawater [4,5,6], fresh water [7], soils [8,9], and multicellular organisms [10,11]. Their ability to infect and kill pathogenic bacteria is employed in clinical, veterinary, food safety, and plant protection practices [12,13,14,15]. On the other hand, tailed phages can disrupt technological processes based on bacterial growth [16,17]. The predator–prey relationships of bacteriophages and bacteria, their genetic interactions resulting in changed prokaryote phenotypes along with phage abundance in nature support fundamental ecological and evolutionary research interests [18,19]. The capacity of a bacteriophage to interact with a certain spectrum of bacterial strains or taxa is described as a host range. Different host range types are defined by the ability of a virus to complete the steps of replication cycle and the biases inherent in laboratory techniques [20]. Existing reviews in the area cover phage infection strategies [21], molecular host range determinants [22], and ecological implications of phage–host interactions [23]. Phage activity spectra spanning multiple bacterial species were discussed in the works by Hyman and Abedon [20] and de Jonge et al. [22]. From a perspective of practical applications, the most important host ranges are those ones manifested by bacterial lysis. The reviews by Casey et al. and Glonti and Pirnay placed lytic host ranges in a broad picture of bacteriophage characteristics relevant to phage therapy [24,25]. The methodological aspects of host range determination were covered by Hyman [26] and Yerushalmy et al. [27].

Along with the type of life cycle, taxonomic affiliation, the absence of bacterial pathogenicity factors in the genome, and the frequency of bacterial resistance, the spectrum of lytic activity is an important characteristic of a therapeutic bacteriophage [12]. The concentration of bacteriophages that target an infectious agent in a preparation influences how well the bacterial population can be controlled [28,29]. Additionally, having a wider host range enables a reduction in the variety of phages used in a cocktail while still preserving a wide range of activity [24,30,31,32]. Discussing particular bacteriophage host ranges, many authors use the terms “narrow” and “broad” despite the apparent absence of quantitative synthesis in the area. Therefore, we addressed the possible presence of a boundary between narrow and broad host ranges in tailed phages taken as a whole. Another research goal was to infer median host ranges and identify significant differences between host ranges of viruses belonging to certain taxonomic groups or infecting particular bacterial species. Also, we tested the hypothesis on fitness cost of broad host ranges and analysed the connection of host ranges with bacteriophage genome size and tRNA number.

Our meta-analysis is based on the proportions of production host isolates that were lysed by particular bacteriophages. For the purpose of this review, we divide the outcomes of the most commonly used methods for determining lytic host range into two categories. Spotting host range reflects the bactericidal effect of a phage preparation. It is assessed in spot tests through placing drops of phage suspension on bacterial lawns or soft agar layers and includes bacterial strains suitable for viral propagation as well as strains susceptible to free lytic components produced during lysate preparation, strains killed through the adsorption of large amount of viral particles and abortive infections. Plaquing host range reflects the spectrum of bacteria supporting phage propagation accompanied by visible destruction of bacterial cells. In the study, we leave aside the question of cross-species specificity and focus on intraspecies host ranges.

## 2. Materials and Methods

### 2.1. Protocol Registration

The initial version of the study protocol was registered at Open Science Framework (DJPFM). Literature search, study identification, and selection complied with the PRISMA guideline [33].

### 2.2. Search Strategy and Eligibility Criteria

A systematic literature search was performed on 17 April 2023 in PubMed and Scopus databases using the terms (((bacteriophage host range isolation) AND (("2011"[Date - Publication] : "2023"[Date - Publication]))) AND (English[Language])) NOT (Review[Publication Type]) for PubMed and the terms TITLE-ABS-KEY ( bacteriophage AND host AND range AND isolation ) AND PUBYEAR > 2010 AND ( LIMIT-TO ( LANGUAGE , "English" ) ) AND ( EXCLUDE ( DOCTYPE , "re" ) ) for Scopus. After de-duplication using PubMed reference numbers and screening for non-original studies, the retrieved full texts were assessed for eligibility (Figure 1, Appendix A). In each study, we evaluated the quality of bacterial strain collections used for host range testing. A collection of at least 10 strains belonging to the same species as production host was considered representative if at least one of the following requirements was satisfied. (1) A typing technique was employed, and there was no particular bacterial type accounting for more than a half of the collection. (2) At least two of the three factors were evidently diversified: sampling location, time on the scale of years, and the type of isolation substrate. (3) The sample of bacterial strains originated from a public culture collection. The preliminary data analysis revealed that the number of plaquing host range observations for *Herelleviridae* and *Drexlerviridae* families was not enough to perform statistical tests. Thus, an additional literature search was carried out, and relevant information from six papers meeting eligibility criteria [34,35,36,37,38,39] was added to the dataset. A total of 282 selected literature sources were included in the reference list [34,35,36,37,38,39,40,41,42,43,44,45,46,47,48,49,50,51,52,53,54,55,56,57,58,59,60,61,62,63,64,65,66,67,68,69,70,71,72,73,74,75,76,77,78,79,80,81,82,83,84,85,86,87,88,89,90,91,92,93,94,95,96,97,98,99,100,101,102,103,104,105,106,107,108,109,110,111,112,113,114,115,116,117,118,119,120,121,122,123,124,125,126,127,128,129,130,131,132,133,134,135,136,137,138,139,140,141,142,143,144,145,146,147,148,149,150,151,152,153,154,155,156,157,158,159,160,161,162,163,164,165,166,167,168,169,170,171,172,173,174,175,176,177,178,179,180,181,182,183,184,185,186,187,188,189,190,191,192,193,194,195,196,197,198,199,200,201,202,203,204,205,206,207,208,209,210,211,212,213,214,215,216,217,218,219,220,221,222,223,224,225,226,227,228,229,230,231,232,233,234,235,236,237,238,239,240,241,242,243,244,245,246,247,248,249,250,251,252,253,254,255,256,257,258,259,260,261,262,263,264,265,266,267,268,269,270,271,272,273,274,275,276,277,278,279,280,281,282,283,284,285,286,287,288,289,290,291,292,293,294,295,296,297,298,299,300,301,302,303,304,305,306,307,308,309,310,311,312,313,314,315].

### 2.3. Data Collection and Interpretation

The information extracted from original sources concerned species identity of bacterial host, bacteriophage genome accession number, morphology of viral particles, method of lytic activity determination, numbers of susceptible and resistant strains, and information about involvement of any approach to uncovering intraspecies diversity in bacterial sample preparation. Bacteriophage names, genome size, and taxonomic information were retrieved from GenBank. The meta-analysis was conducted with data fitting broad descriptions of spotting and plaquing host ranges. The principal difference between the two was considered to be proven ability of a phage to reproduce on a particular bacterial strain. In our dataset, spotting host range included the results of spot tests as well as the cases where positive testing results indiscriminately included lysis spots and plaques, e.g., [45,187,219]. This category included experimental results obtained with or without phage stock purification either on bacterial lawns or agar overlays. We classified all spot test results as positive or negative, including turbid spots or incomplete clearing in the count of positive outcomes. Plaquing host ranges were calculated from results associated with plaque observations. In some papers, spot test results were not reported independently of plaque-based assay results, e.g., [160,199]. In those cases, host ranges were considered to be plaquing. Production strains were included in host range calculations. Genetically modified bacterial strains were not considered, since, in most cases, the manipulations were performed in order to identify phage receptors [192] or address other aspects of bacteriophage life cycle [64]. Initially, this study was designed to include non-tailed phages and the results obtained with liquid culture-based methods. But we found only one non-tailed phage in the data and limited the analysis to *Caudoviricetes*. Liquid culture-based methods were used for host range estimation in 2% of papers, and the related data were not collected. When information on efficiency of plating (EOP) was available, we recorded the outcomes in three categories. These included phage-resistant strains, strains with EOP values below 0.1, and strains with EOP values above or equal to 0.1. The use of EOP categories allowed us to include into analysis both uncategorised and categorised data. Highly deviating EOP estimates from two publications were removed from the dataset [157,239].

### 2.4. Taxonomic Analysis

Taxonomic data were downloaded from the NCBI databases Nucleotide and Taxonomy (https://www.ncbi.nlm.nih.gov/, accessed on 12 February 2024). To increase the number of bacteriophage groups available for statistical analysis, whole-genome sequences of the bacteriophages described in the selected literature were annotated with the use of Prokka 1.14.6 [316] and employed in phylogenetic network analysis in vContact2 0.11.3 [317], with subsequent visualisation in Cytoscape 3.10.1 [318]. The parts of the phylogenetic network were subjected to phylogenomic analysis on ViPTree web service (https://www.genome.jp/viptree/, accessed on 15 February 2024). The family-level groups (FLGs) were assigned after searches for shared protein-coding genes using CoreGenes 5.0 web server and Bidirectional Best Hit group-clustering algorithm (https://coregenes.ngrok.io/, accessed on 19 February 2024) [319,320]. The shared gene analysis was carried out with annotations available in GenBank. The bacteriophage life cycles were predicted by PhaTYP script as implemented in PhaBOX tool set [321,322]. Fitness cost analysis took into account predictions with a score of 0.95 or higher. The life cycles of *Aliceevansviridae* were set to temperate manually [323].

### 2.5. Statistical Analysis

The distribution of host ranges in the general sample of 908 bacteriophages was analysed in Python libraries SciPy v1.11.1 [324] and statsmodels v0.14.0 [325]. Uniform distribution was modelled with the use of cumulative distribution function *F*(*x*) = *x* in the [0, 1] interval, and a mixture of uniform and triangular distributions was modelled as *F*(*x*) = 1.5 × *x* − 0.5 × *x*^2^ in the [0, 1] interval. The comparative analyses were performed with phage families and bacterial species using R v4.3.3 [326]. The host range values were tested for differences with the use of Mann–Whitney non-parametric test at 95% confidence interval. The required size of every particular group to be included in the statistical analysis was set at 15.

Sources of heterogeneity in the data and the relationships between variables were identified by factor analysis of mixed data (FAMD) in the R package FactoMineR version 2.11 [327]. We selected four virus–host pairs with the highest numbers of associated publications, including *Drexlerviridae* and *Klebsiella pneumoniae*, *Straboviridae* and *Escherichia coli*, *Guernseyvirinae* (referred to as FLG-G) and *Salmonella enterica*, and *Herelleviridae* phages tested with *Staphylococcus aureus* strains (Appendix A). The categorical variables were virus–host pairs, employment of host typing techniques, and culture medium. The continuous variables were the number of predicted tRNA genes in the bacteriophage genome, the number of bacteriophage particles per spot, and host range. The tRNA-coding gene prediction was performed with the use of Prokka 1.14.6. Data standardisation was carried out via the built-in algorithm of FactoMineR and included scaling continuous variables to unit variance and transformation of the categorical variables into a disjunctive data table with subsequent scaling using the specific scaling of multiple correspondence analysis. The R scripts for data formatting and visualisation were included into Appendix A.

## 3. Results and Discussion

### 3.1. Dataset

The data extracted from 282 eligible papers were related to 908 unique tailed bacteriophages and included 565 spotting host range observations and 439 plaquing host range observations. For 11% phages, paired host range data points were available (*n* = 96). Due to low proportion of paired data points, the two types of host range data were analysed separately. Taxonomic annotations at the levels of families, subfamilies, and genera were available for 53%, 46%, and 82% of phage genomes, respectively. For 99 phages, there was no taxonomic annotation below order *Caudoviricetes* (11%). Host genus and genome size distributions were similar to those ones in the reference database of phage genomes INPHARED. Cook et al. described the database as being dominated by phages that infect a small number of bacterial genera [328]. The same applied to our dataset with 40 host genera, with 77% consisting of *Escherichia*, *Salmonella*, *Pseudomonas*, *Klebsiella*, *Staphylococcus*, *Flavobacterium*, *Streptococcus*, *Yersinia,* and *Lactococcus* phages (Figure 2a). *Mycobacterium* phages accounted for the major difference between the datasets, being the most prevalent group in INPHARED, but completely absent from our dataset. This difference can be explained by genomic focus of the project devoted to *Mycobacterium* phage description [328,329]. The bacteriophage genome size varied from 17.5 to 370.8 kb, with prominent peaks at 35–50 and 170 kb. The 35–50 kb peak was populated by all three morphotypes, and the 170 kb peak was constituted by myoviruses (Figure 2b). The observed peak pattern was very close to what was described for the INPHARED database, though the genome size distribution of our dataset lacked the 5–10 kb peak associated with non-tailed *Microviridae* phages.

### 3.2. No Clear Boundaries Between Narrow and Broad Host Ranges

To address the question of broad and narrow host ranges in tailed phages taken as a whole, we visualised the host range distributions. An evident area with relatively rare values dividing a distribution would substantiate a distinction between the two. However, there were no apparent breaks. Most peaks were formed by published series of bacteriophages with similar host ranges, implying a publication bias (Figure 3a,b).

To assess the true form of the distributions in the general population, we overlaid 500 resample histograms. Every generated resample of host range values comprised one randomly chosen measurement for each literature source. The resulting graph displayed less prominent peaks. The overlaid visualisation of spotting host range (SHR) resamples suggested the uniform distribution (a horizontal line) on the interval [0, 1] (Figure 3c). Overlaid plaquing host range (PHR) value distributions display similarity to a mixture of the uniform and triangular distributions on [0, 1] with mode at 0 (a down-sloping line) (Figure 3d). In order to quantify this similarity, one-sample Kolmogorov–Smirnov (KS) and Cramer–von Mises (CvM) tests were performed on a total of 10^6^ resamples (out of 8 × 10^43^ possible combinations). Both the KS test and CvM criterion are goodness-of-fit tests for cumulative distribution function given an empirical distribution. They allow us to estimate the probability of a sample coming from a given distribution, uniform for SHR and equal mixture of uniform and triangular, with probability density *p*(*x*) = 1.5 − *x*, for PHR. The *p*-values of KS test and CvM criterion were considered as random variables. For SHR, the *p*-values of the KS and CvM tests were 0.37 ± 0.15 and 0.23 ± 0.11, respectively (mean ± SD, *n* = 10^6^). Hence, the hypothesis that SHR distribution is uniform was not rejected. The inferred true distribution of PHR values was indistinguishable from the modelled mixture of uniform and triangular distributions with KS *p*-value at 0.88 ± 0.12 and CvM *p*-value at 0.89 ± 0.10 (mean ± SD, *n* = 10^6^). Since uniform and triangular distributions are indivisible, we could not discriminate between narrow and broad host ranges in *Caudoviricetes* taken as a whole.

### 3.3. Family-Level Groups of Phages

To increase the coverage of taxonomic annotation, we (1) identified non-assigned members of major phage families and (2) evaluated subfamilies as possible substitutes for families in statistical analysis where the taxa accepted by International Committee on Taxonomy of Viruses (ICTV) were polyphyletic or lacking. Both approaches relied on the results of a phylogenetic network analysis (Figure 4). The major parts of the phylogenetic network containing annotation inconsistencies were analysed using a phylogenetic tree construction. It allowed for adding family annotations to one *Drexlerviridae* member (Appendix A), three *Straboviridae* members (Appendix A), and three *Autographiviridae* members (Appendix A). The group of *Aliceevansviridae* phages remained unchanged after the phylogenetic tree analysis (Appendix A).

In the taxonomy-based analysis of host ranges, we regarded *Guernseyvirinae* as a family and divided *Autographiviridae* in two parts. Currently, the subfamily *Guernseyvirinae* is an independent taxon within *Caudoviricetes*. According to ICTV, a viral family should be represented by a cohesive and monophyletic group inferred by standard proteome-based clustering tools. Its members are expected to share approximately 10% of orthologous genes, depending on the genome size and number of coding sequences [1]. *Guernseyvirinae* phages in our dataset shared six genes (*n* = 36). This number was very close to a required threshold of 10%, taking into account that the number of protein-coding genes in the group varied between 44 and 75. Thus, we included *Guernseyvirinae* in the host range analysis as a “family-level group consisting of *Guernseyvirinae* phages”, or FLG-G for short. The members of *Autographiviridae* formed two isolated parts of the phylogenetic network. One part comprised the members of subfamily *Studiervirinae* with ten shared genes (inferred from 49 genomes), and the other part comprised the rest of *Autographiviridae,* sharing four common genes (*n* = 82). The combined sample of *Autographiviridae* genomes (*n* = 131) shared two genes. Consequently, we considered *Studiervirinae* as FLG-AS. Phylogenetic analysis did not challenge its monophyly (Appendix A). The rest of *Autographiviridae* were taken into host range analysis as FLG-A.

In addition to the annotation inconsistencies, there were two edge cases of phages residing within a cohesive part of the phylogenetic network with the members of major phage taxa and branching at the roots of these taxa on the phylogenomic trees. In the CoreGenes analysis, the unification of Escherichia phage vB_EcoS_PTXU06 (MK373789), Escherichia phage Gluttony_ev152 (LR597646), and Escherichia phage vB_EcoS_WF5505 (MK373790) with FLG-G led to a reduction in the number of shared genes from six to two. These three phages were not considered in the host range analysis of bacteriophage taxa. Rhizobium phage RHEph01 (JX483873, *Autographiviridae*) resided within the *Studiervirinae* part of phylogenetic network but clustered as an outgroup to the core *Studiervirinae* members. The addition of RHEph01 genome to *Studiervirinae* phages from our data decreased the number of shared protein-coding genes from ten to seven. Given that the genomes of core *Studiervirinae* members coded for 43–58 proteins, we included RHEph01 in the FLG-AS dataset.

### 3.4. Host Ranges Differ Between Taxonomic Groups

To a certain extent, taxonomic identity of a bacteriophage can provide information about its ability to lyse wider or narrower spectrum of host isolates. On the one hand, the host ranges within viral families varied noticeably (Figure 5a,b), but in some cases the differences between the groups were significant (Figure 5c,d). *Aliceevansviridae* was the only group of temperate bacteriophages in the analysis. It is to be expected that in host range tests a part of infecting virions entered lysogenic cycle, thereby limiting observed lytic activity spectra. The two parts of *Autographiviridae* FLG-A and FLG-AS had statistically different plaquing host ranges (Figure 5d). To obtain an insight into this difference, we focused on two shared bacterial hosts. In *Klebsiella pneumoniae*, nearly all *Autographiviridae* phages were specific to one capsular type [184,233,268,275], irrespective of their phylogenetic position. The only exception was one FLG-AS phage able to propagate on three capsular types of *K. pneumoniae* [148]. The majority of host range data concerning *Pseudomonas syringae* lysis by *Autographiviridae* phages were obtained with one or two bacterial pathovars. In both groups, there were phages able to propagate on all tested pathovars [183,252]. In part, the difference between PHR of FLG-A and FLG-AS can be explained by a published series of eight FLG-A phages infecting *A. baumannii* [206]. In the publication, host range testing of capsular-type-specific phages was performed with 56 *A. baumannii* strains possessing different capsular types, resulting in PHR values close to zero. As of the time of writing, GenBank does not contain genomes of FLG-AS representatives specific to *A. baumannii*. Therefore, there may be biological peculiarities underlying differences in host species specificity.

In the FLG-G and *Herelleviridae* data, there was one prevailing bacterial host, represented by *Salmonella enterica* and *Staphylococcus aureus*, respectively. The differences between host ranges between the bacteriophage groups can be explained by the structure of bacterial cell wall and the biology of virus–host interactions. While FLG-G phages recognise polymorphic serotype-specific *Salmonella* O-antigen [139,330], *Herelleviridae* members adsorb to the conservative backbone of *Staphylococcus* wall teichoic acid [331]. In our host range analysis of bacteriophage family-level taxonomic groups, median host ranges of *Aliceevansviridae* and *Herelleviridae* phages represented minimum and maximum values. Leaving these two groups aside, the only pair of FLGs with significant differences between both types of host ranges were podoviruses FLG-A and myoviruses *Straboviridae* (Figure 5c,d).

Spotting and plaquing host ranges of the same phage may differ. For the study, 96 paired host range assessments were available. The detectable differences between spotting and plaquing host ranges (>0.1, Appendix A) were found in 29% paired data points. In 6% of cases, the difference between host ranges exceeded 0.5. However, multiple studies of *Drexlerviridae* members [38,150,221,284,309] and phages infecting *Vibrio parahaemolyticus* [190,209,267,298] repeatedly reported equal spotting and plaquing host ranges. On the other hand, all available for the study paired spotting and plaquing host ranges of *Herelleviridae* differed [34,59,112], probably due to the high activity of phage endolysins [332].

Our analysis provides a framework for discussing phage therapy strategies. The effective clearance of infection site requires either high phage concentration or high numbers of susceptible bacteria [333], and low titres of active phage in a preparation may cause treatment to fail [334]. With other practice-relevant phage properties like adsorption rate, latent period, burst size, and virion half-life being equal [335,336], broad host range is a desirable trait. The structure of infection networks suggests that generalist phages tend to succeed in lysis of bacteria infected by lower numbers of different phages [337]. Thus, the ease of usage and universality of ready-to-use preparations are achievable through combining broad host range bacteriophages and phage cocktails probably cannot be seen as a means of expanding the host range through combination of highly specific individual viruses.

In our dataset, the phages active against *Staphylococcus aureus* and *Pseudomonas aeruginosa* had the broadest host ranges (Figure 6a,b). The two species were the targets of the three ready-to-use phage preparations with efficacy demonstrated in clinical trials. The other known ready-to-use preparations with unconfirmed efficacy targeted *Proteus mirabilis*, *Enterococcus* spp. and *Escherichia coli* [338,339]. Therefore, application areas of ready-to-use preparations and personalised treatment of infections can be delineated on the basis of host range breadth.

The inferred median bacteriophage host ranges in relation to particular bacterial species confirm general estimates that host ranges in most cases do not cover more than 40–50% of bacterial strains [340]. Particularly, in our study, the median host ranges of *K. pneumoniae* phages were estimated at 21–25% of bacterial strains. It was perfectly in line with an observation that the majority of *Klebsiella* phages infect 9–30% of the tested strains, while the minority of these viruses show relatively broad lytic spectra at 35–50% [341].

Broad host ranges come at a fitness cost but probably not always. We hypothesised that in strictly lytic phages, the ability to lyse wider spectrum of production host isolates comes along with decreased efficiency of plating. To test it, we plotted the ratio of strains lysed at EOP values above or equal to 0.1 to the total number of successful infections against plaquing host ranges (Figure 7). The possible association was estimated by fitting a linear regression model to the three parts of data, including the phages of the two most represented species *Salmonella enterica* (*n* = 12, *nls* = 8) and *Escherichia coli* (*n* = 9, *nls* = 5) and phages infecting other bacterial species (*n* = 70, *nls* = 32). Broader lytic spectra of *Escherichia* phages were associated with lower proportions of efficiently utilised host strains. Earlier, Maffei et al. studied an original sample of 68 *Escherichia* phages and found similar trade-offs between broad host recognition and plaquing host range [342]. Our coliphage EOP data did not include the results of Maffei et al. and were compiled from five other publications [115,142,164,232,290], providing an independent confirmation. However, we could not see these trade-offs in a broader spectrum of bacteriophage hosts. Therefore, further experimental research on well-defined bacteriophage and bacterial collections should clarify the issue.

### 3.5. Taxonomy of Phages and Hosts Was the Only Identified Factor, Correlating with Host Ranges

There are sources comparing host ranges in relation to bacteriophage morphotypes. In a study of marine phages, myoviruses had broader host ranges than siphoviruses [343]. In other studies, podoviruses were found to be inferior to other bacteriophage morphotypes in terms of spotting and plaquing host ranges [145,344,345]. In agreement with an experimental paper questioning this rule [346], we did not observe a connection between the host ranges and phage morphology (Figure 8).

In tailed bacteriophages, there is a positive association between genome size and the number of tRNAs, which are considered to be a viral adaptation factor [347,348]. Larger genomes have a capacity to harbour phage counter-defence systems and enzymatic activities that enable weathering host immune attacks and overcoming metabolic restrictions, thereby defining an evolutionary strategy of stronger competitors [349]. Therefore, we hypothesised that bacteriophages with larger genome size have broader intraspecies host ranges. However, this was not the case (Figure 8a,b). One possible explanation can come from the tendency of phages with larger genomes to be commonly targeted by bacterial protective systems, triggering an altruistic suicide of infected cells [342].

The study of Delesalle et al. focused on tRNA gene distribution in mycobacteriophages. They explained the presence of tRNA genes in bacteriophage genomes by differences in prevalence of particular amino acids in phage and host genomes. Since closely related phages possessing different numbers of tRNA genes infect common bacterial hosts, the higher numbers of tRNA genes hypothetically may be associated with broader lytic spectra [350]. To find if the number of predicted tRNA genes, bacteriophage and host taxonomy, culture medium type, and availability of host typing data correlate with host range values, we performed a factor analysis of mixed data. In SHR data, virus–host pairs contributed most to the second dimension. Most contributions to the first dimension came from virus–host pairs and spotting-host ranges, implying a correlation between the two (Figure 9a). In PHR data, the most contributing variables were culture medium, virus–host pairs, PHR, and the number of tRNA-coding genes. The PHR variable correlated with culture medium type and virus–host pairs (Figure 9b). However, the correlation between the PHR variable and culture medium was due to the use of different culturing media for different bacteria. Taken together, these results suggest virus and host taxonomy is a key factor determining the host ranges.

### 3.6. Methodological Limitations and Reliability of the Study

The limitations of our study are connected to (1) sample representativeness, (2) host range variability, and (3) methodological issues. Metagenomic data indicate higher diversity of non-cultivable viruses in comparison to the number of described bacteriophages [351]. Also, phage isolation techniques tend to be selective towards particular taxonomic groups of phages [352]. Our review concerns cultured phages, and, inevitably, the conclusions cannot be extrapolated to the diversity of tailed bacteriophages in general. Moreover, selective reporting is the common practice in the description of bacteriophages intended for the use in biocontrol applications. The list of sources explicitly stating pre-selection of broad host range viruses may include the references [75,76,172,199,287,291].

The host ranges are not constant and can be manipulated in vitro [340]. They change due to mutational processes in bacteria and viruses and emergence of phenotypic resistance mechanisms in bacteria [353]. The temporal changes in host ranges of the bacteriophages can be observed in marine environment [261], industrial and clinical settings [111,354]. However, the distribution of median bacteriophage host ranges is tractable in terms of bacteriophage life cycles and phage–host interactions. In some aspects, our key observations were confirmed by large-scale experimental studies.

Abiotic factors influence bacteriophage activity [355,356]. In some cases, methodological details omitted here for the sake of study feasibility were shown to influence the host range estimates. The addition of Ca^2+^ and Mg^2+^ ions can stabilise the interaction of the virion with host cell [94,357,358]. This stabilisation may determine the test outcome [253]. Host range breadth depends on incubation temperature [253,283,359] and time [87]. Still, our approach to study selection in relation to bacterial sample quality was robust. It can be seen from the negligible contributions of the host typing employment factor to the data variability. In the view of high diversity of host range determination techniques, the classification of host ranges may not be very precise. Therefore, we expect that real-world differences between host ranges of different bacteriophage taxa are more pronounced in comparison to what can be seen in our data.

## 4. Questions for Further Studies

The statement on “high bacteriophage specificity” is closely followed by a question about the degree of this specificity. The approach employed here answered this question for several well-studied taxonomic groups of bacteriophages and bacteria. Future studies may involve other taxa in similar analyses. Leaving *E. coli* phages aside, is there fitness cost of broad host ranges? If so, is it relevant to phage biocontrol applications? Finally, our study clearly revealed a need for improvements in bacteriophage macrotaxonomy.

## 5. Conclusions

In tailed bacteriophages, published lytic host ranges estimated on production host isolates vary from close to 0 to 1. There is no boundary between narrow and broad host ranges. In some cases, host range differences between bacteriophage families are significant. The same applies to phages, grouped by bacterial host species. We showed that in most lytic phages, broad host ranges come at no fitness cost in terms of efficient host utilisation and rejected the hypotheses about a connection of host ranges with bacteriophage genome size and tRNA number. In this study, the only revealed factor correlating with the breadth of lytic spectra was bacteriophage and host taxonomy. Though bacteriophage susceptibility testing by standard methods is a cornerstone of phage biocontrol applications and our approach had a number of limitations, we believe our review adds to understanding of phage functional diversity.

## Figures and Tables

**Figure 1 viruses-16-01879-f001:**
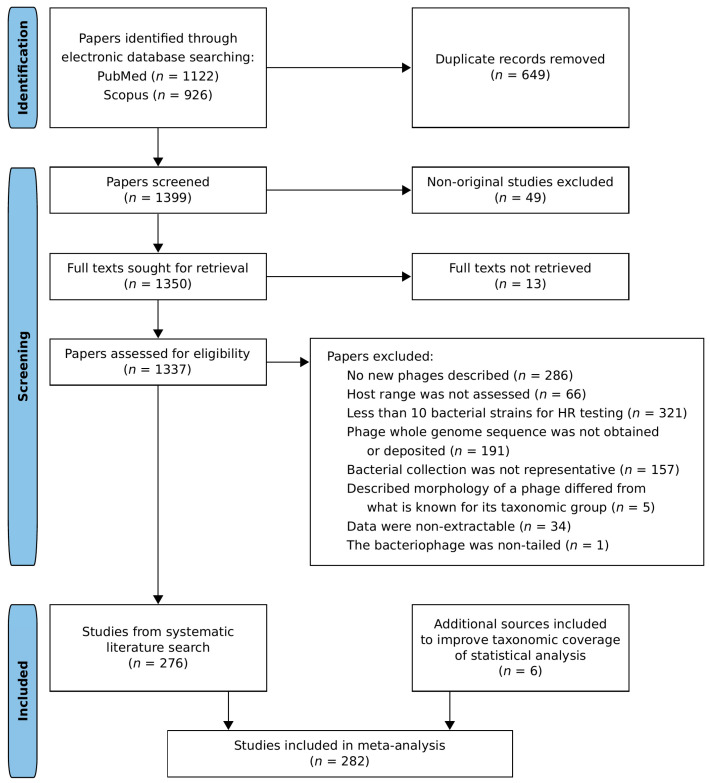
PRISMA flowchart of source identification and selection. PRISMA, preferred reporting items for systematic reviews and meta-analyses [33].

**Figure 2 viruses-16-01879-f002:**
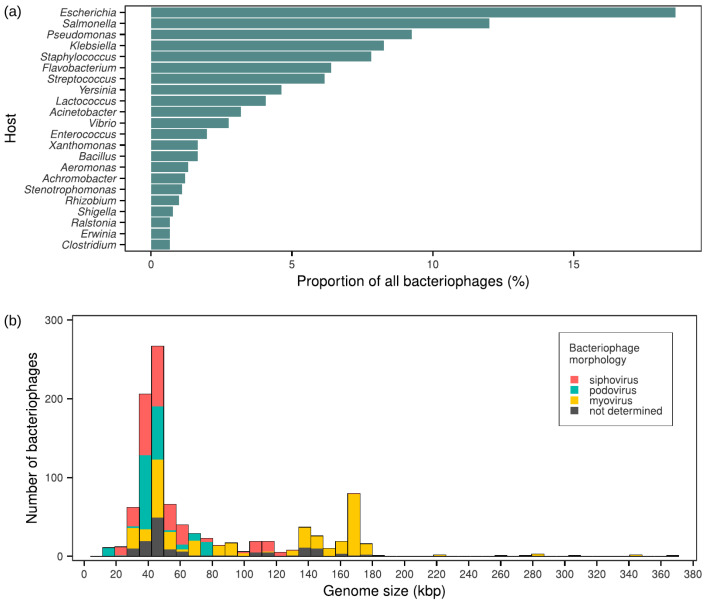
Overview of the dataset: (**a**) bacterial host genera arranged by their prevalence (top 22 genera are shown); (**b**) distribution of bacteriophage genome size and morphology.

**Figure 3 viruses-16-01879-f003:**
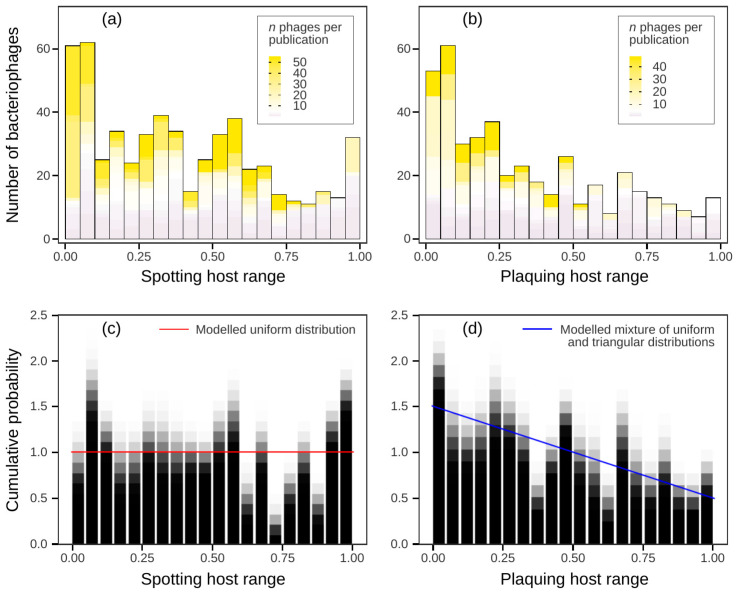
Host range distributions are biased by published phage series and cannot be divided in parts. (**a**,**b**) The peaks in the original distributions are formed by data points originating from papers with high numbers of described bacteriophages, uncovering the prevalence of published series of bacteriophages with similar host ranges and a publication bias of large datasets. (**c**,**d**) The unbiased forms of the distributions were inferred by random choice of one data point from each paper in 500 replicates. The superimposed visualisations implied uniform distribution of spotting host ranges and a mixture of uniform and triangular distributions for plaquing host ranges. The visualisations are based on the host range data of all selected tailed bacteriophages, including the viruses with unknown taxonomic position within the class *Caudoviricetes*.

**Figure 4 viruses-16-01879-f004:**
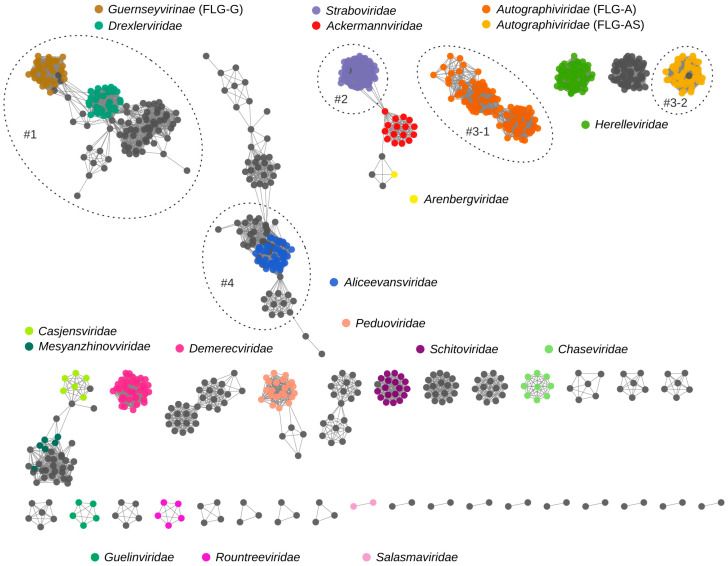
Genetic diversity of bacteriophages in the sample visualised by phylogenetic network. The groups of phage genomes numbered from #1 to #4 were further analysed by phylogenetic tree construction (Appendix A). Subfamily *Guernseyvirinae* was included in host range data analysis as a family-level group FLG-G. Host ranges of *Autographiviridae* members were analysed in two independent groups, FLG-A and FLG-AS (*Studiervirinae*).

**Figure 5 viruses-16-01879-f005:**
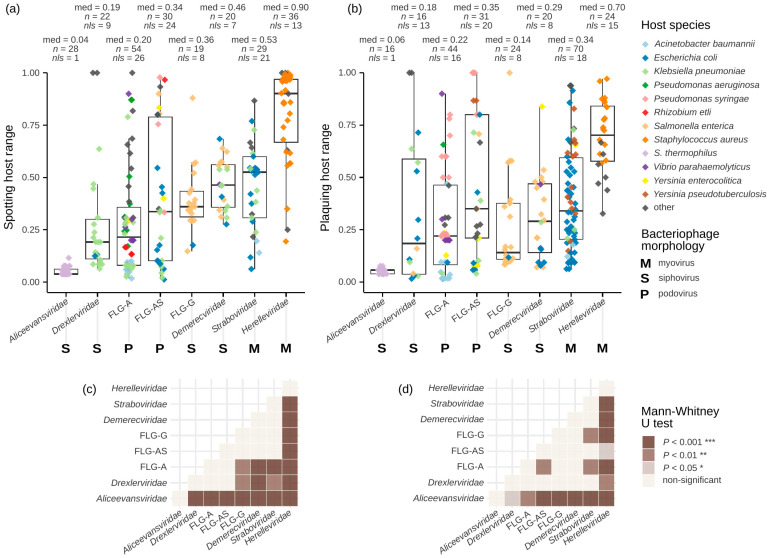
The distributions of bacteriophage host ranges differed between phage family-level groups: (**a**) Spotting host ranges. (**b**) Plaquing host ranges. Given the estimated error of repeated host range assessments at 0.1 (Appendix A), PHR of *Straboviridae* and FLG-G, and the two host range types in *Drexlerviridae*, FLG-A and FLG-AS varied across the entire interval of values. (**c**) Differences between SHR value distributions and (**d**) differences between PHR value distributions assessed by Mann–Whitney U test. The order of family-level taxonomic groups of phages follows the increase in median spotting host ranges. med, median; *n*, number of host range data points; *nls*, number of literature sources; *S. thermophilus*, *Streptococcus thermophilus*.

**Figure 6 viruses-16-01879-f006:**
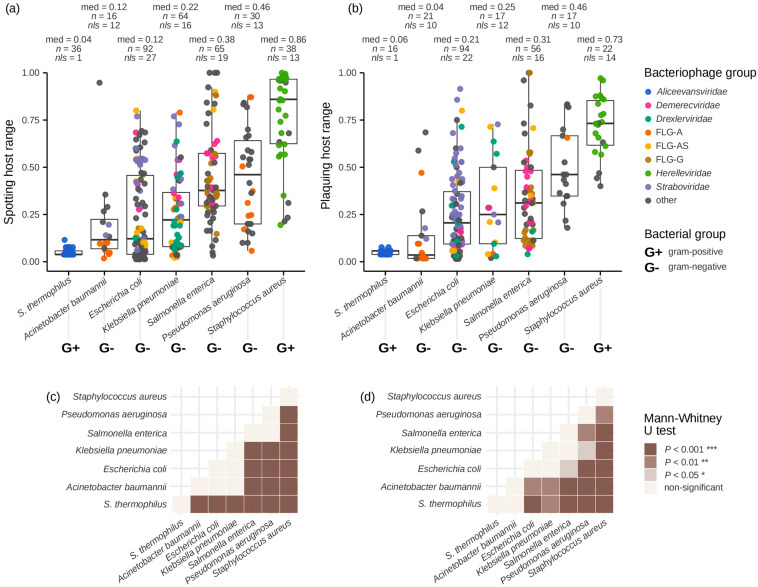
The distributions of bacteriophage host ranges differed between bacterial species: (**a**) Spotting host ranges. (**b**) Plaquing host ranges. Given the estimated error of repeated host range assessments at 0.1 (Appendix A), SHR of *A. baumannii* phages, PHR of *E. coli* phages, and the two host range types of *S. enterica* phages varied across the entire interval of values. (**c**) Differences between SHR value distributions and (**d**) differences between PHR value distributions assessed by Mann–Whitney U test. The order of bacterial species follows the increase in median spotting host ranges. med, median; *n*, number of host range data points; *nls*, number of literature sources; *S. thermophilus*, *Streptococcus thermophilus*.

**Figure 7 viruses-16-01879-f007:**
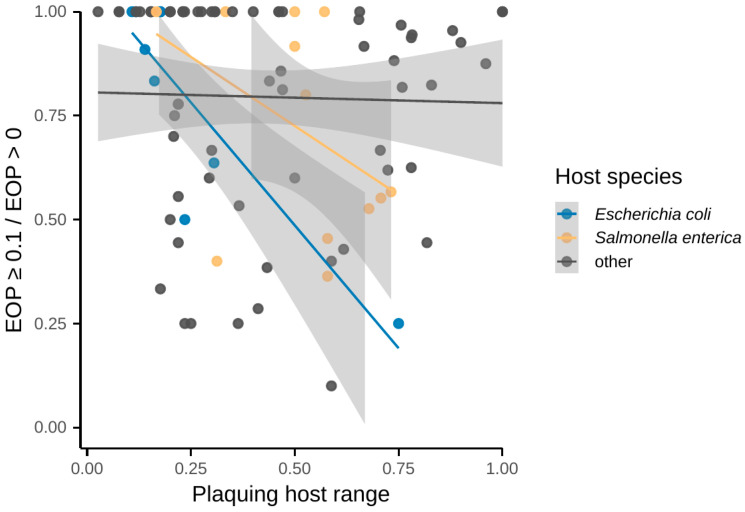
In strictly lytic bacteriophages, broader plaquing host ranges may be associated with decreased proportion of efficiently utilised host strains. The lines depict linear regression analysis results with 95% confidence intervals shaded grey. In *E. coli* bacteriophages, there is a negative correlation between plaquing host ranges and the efficiency of plating (EOP). In the groups of bacteriophages propagating on *S. enterica* and all other hosts, the correlation cannot be seen.

**Figure 8 viruses-16-01879-f008:**
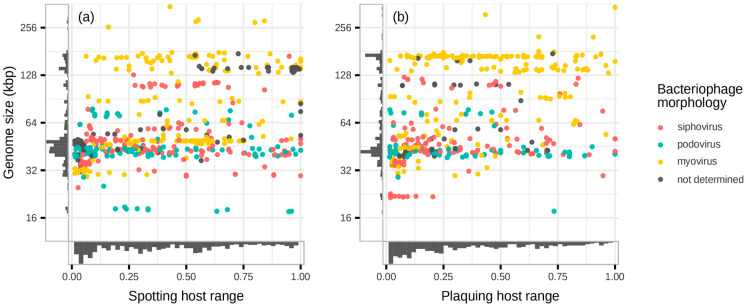
Bacteriophages morphotypes in the coordinates of host range and genome size: (**a**) Spotting host range. (**b**) Plaquing host range. There is no correlation between host ranges, morphotypes, and genome size. The visualisations are based on host range data of all selected tailed bacteriophages, including the viruses with unknown taxonomic position within the class *Caudoviricetes*.

**Figure 9 viruses-16-01879-f009:**
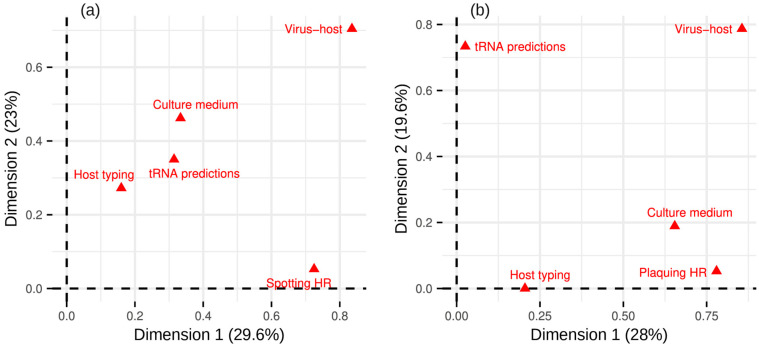
Sources of heterogeneity identified by factor analysis of mixed data: (**a**) Spotting host ranges. (**b**) Plaquing host ranges.

## Data Availability

The dataset and code are available as Appendix A to the article. The study was registered at Open Science Framework website (https://osf.io/djpfm, accessed on 3 December 2024, DOI: 10.17605/OSF.IO/DJPFM).

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
