# Peer review of "Lytic Spectra of Tailed Bacteriophages: A Systematic Review and Meta-Analysis"

_viruses, 2024, doi:10.3390/v16121879_

Round 1
Reviewer 1 Report
Comments and Suggestions for Authors
In this manuscript, the authors analyzed the host range of tailed phages from published data. They Calculated the host range values as a ratio of lysed strains to the number of tested strains.
However, this can be affected by number of strains tested in each paper, taxonomic group of tested bacteria. For example, if large numbers of related bacteria or different strains of the same species were tested, the value will close 1 but if they belong to different taxonomic group, the value will close 0. Therefore, “narrow” or “wide” host range can be determined by this analysis. Host range can be defined by the infection ratio in different strains of the same species or different species only depend on the purpose of study. Therefore, the definition of narrow and wide can be subjective, which could not be concluded by this type analysis.
Author Response
In this manuscript, the authors analyzed the host range of tailed phages from published data. They calculated the host range values as a ratio of lysed strains to the number of tested strains.
However, this can be affected by number of strains tested in each paper (Comment 1), taxonomic group of tested bacteria (Comment 2). For example, if large numbers of related bacteria or different strains of the same species were tested, the value will close 1 but if they belong to different taxonomic group, the value will close 0. Therefore, “narrow” or “wide” host range can be determined by this analysis (Comment 3). Host range can be defined by the infection ratio in different strains of the same species or different species only depend on the purpose of study (Comment 4). Therefore, the definition of narrow and wide can be subjective, which could not be concluded by this type analysis (Comment 5).
Response 1. Dear Reviewer, thank you indeed for the helpful criticism. To find, if the host range values were affected by number of bacterial strains tested, we amended the R code previously used for generation of Figures 5 and 6 to include information about the bacterial sample size. The result can be seen in the four upper panels of the illustration. There is no apparent dependence of host range medians on sample size. In Klebsiella pneumoniae phages, the narrowest plaquing host ranges were obtained on one of largest samples. The largest bacterial sample to test spotting host ranges of FLG-AS phages can be seen very close to median SHR value. To prepare the two lower panels, we plotted host range data available for the study against bacterial sample size. No linear or any other dependence of host ranges on sample size is recognisable. Through the supplementary file Data S1 this information is traceable back to the original literature sources.
Response 2. Please kindly refer to our conclusions. Indeed, we found that host range values differ between bacteriophages infecting different bacterial species.
Response 3. As can be judged upon available data, broader host ranges are not associated with larger bacterial strain samples. Please also see our Response 1 and the section “Search strategy and eligibility criteria”.
Response 4. We fully agree that host range can be defined by the infection ratio in different strains of the same species or different species only depending on the purpose of study. We are also aware of literature not recommending to report host ranges as decimals. However, the very goal of the study was to compare host range data from the viewpoint of modern phage and bacterial taxonomy.
Response 5. We admit that the definition of narrow and wide can be subjective. For example, one can compare phages infecting particular bacteria or belonging to some taxonomic groups. Our subjective definition explicitly relates to tailed phages taken as a whole, and we humbly believe the conducted data retrieval and analysis will help other researchers to make their own comparisons.
Are the methods adequately described? (Must be improved, Comment 6).
Response 6. As no specific changes were requested, we improved method description by expanding the explanation of host range value calculation.
Are the results clearly presented? (Must be improved, Comment 7).
Response 7. The captions of Figures 5 and 6 were improved for clarity. We also introduced a number of corrections throughout the text.

Reviewer 2 Report
Comments and Suggestions for Authors
The authors performed a statistic analysis on the factors determining the host ranges that the tailed phages can exert their effects upon. The approaches applied were multidimensional and the results were clearly presented. The authors found that the taxonomy may be the most reliable cue for the estimation of a phage's host range so far. I believe this is a scientifically solid paper that would not only help the experts to gain an universal view on this topic, but also inspire those who do not work in this field. I appreciate the authors' tremendous efforts in building this work and I suggest this manuscript to be published as is.
Author Response
The authors performed a statistic analysis on the factors determining the host ranges that the tailed phages can exert their effects upon. The approaches applied were multidimensional and the results were clearly presented. The authors found that the taxonomy may be the most reliable cue for the estimation of a phage's host range so far. I believe this is a scientifically solid paper that would not only help the experts to gain an universal view on this topic, but also inspire those who do not work in this field. I appreciate the authors' tremendous efforts in building this work and I suggest this manuscript to be published as is (Comment 1).
Response 1. Thank you very much for the positive comment.
Reviewer 3 Report
Comments and Suggestions for Authors
The manuscript is an interesting review aimed at presenting in a very up-to-date and adequate way the topic of the lytic spectrum of tailed phages. The topic is in itself quite complicated to present for several reasons, among which we mention the enormous genetic variability of phages, which makes it difficult to classify them, the diversity of methods to determine lysis of sensitive bacteria, the number of different phages for different species, etc. An intensive bibliographic search was carried out and it was processed in different ways to be able to show the clearest possible results. The authors recognize that the study carried out has limitations and it is correct that they express it. I have no objections regarding the content of the writing but I point out some things related to the language (see Quality of English...)
Comments on the Quality of English Language- Review the use of italics in References for genus, species, families....
- Standardize the way of citing other authors in the text. In some places "et al." is used. and in others "....and coauthors"
- lines 268- 271: italics for the genus
Author Response
The manuscript is an interesting review aimed at presenting in a very up-to-date and adequate way the topic of the lytic spectrum of tailed phages. The topic is in itself quite complicated to present for several reasons, among which we mention the enormous genetic variability of phages, which makes it difficult to classify them, the diversity of methods to determine lysis of sensitive bacteria, the number of different phages for different species, etc. An intensive bibliographic search was carried out and it was processed in different ways to be able to show the clearest possible results. The authors recognize that the study carried out has limitations and it is correct that they express it. I have no objections regarding the content of the writing but I point out some things related to the language (see Quality of English...)
Comments on the Quality of English Language
- Review the use of italics in References for genus, species, families… (Comment 1).
Response 1. Thank you for reviewing the manuscript. We corrected spelling of taxonomic names in References.
- Standardize the way of citing other authors in the text. In some places "et al." is used. and in others "....and coauthors" (Comment 2)
Response 2. We substituted "....and coauthors" for "et al.".
- lines 268-271: italics for the genus (Comment 3)
Response 3. On the lines 268-271 of the original version, genus names constitute parts of bacteriophage names. Therefore, upright spelling is correct. For this matter, please kindly refer to the document “How to write virus, species, and other taxa names” at the official website of International Committee on Taxonomy of Viruses (https://ictv.global/files/info, accessed on 19 November 2024).
Round 2
Reviewer 1 Report
Comments and Suggestions for Authors
By revision of the original version, most of the questions have been cleared. The results showed that there is some tendency of host range among the bacteriophage taxonomy groups. However, the results presented in this manuscript can be affected by the definition of “Host range”. Depend on purpose, many strains of the same species from which the bacteriophage had been isolated can be tested or many different genus or species can be tested. Therefore, the authors need to clarify this aspect in introduction, material methods. Also, the data need to be analyzed depend on these two cases, which can not be differentiated present manuscript.
Author Response
Dear Reviewer, thank you again for your comments. Please find our responses below.
R. By revision of the original version, most of the questions have been cleared. The results showed that there is some tendency of host range among the bacteriophage taxonomy groups. However, the results presented in this manuscript can be affected by the definition of “Host range”. Depend on purpose, many strains of the same species from which the bacteriophage had been isolated can be tested or many different genus or species can be tested. Therefore, the authors need to clarify this aspect in introduction, material methods (Comment 1).
Response 1. The general definition of "Host range" can be found at the Lines 37-40 (Introduction). The working definitions of spotting and plaquing host ranges were given at the Lines 64-74 (Introduction) and the detailed explanation of what was done to prepare the data set exists as the Section 2.3."Data collection and interpretation". We admit that there is a need for highlighting the matter in the Materials and Methods and refine the explanation of eligibility criteria at the lines 88-90.
R. Also, the data need to be analyzed depend on these two cases, which can not be differentiated present manuscript (Comment 2).
Response 2. The host ranges spanning different bacterial species and genera were beyond the scope of the study (Lines 13-14 and 23-24, Abstract; Lines 64-65 and 73-74, Introduction; Line 467, Conclusions).
Reviewer 3 Report
Comments and Suggestions for Authors
The authors have responded satisfactorily to the comments made and the manuscript appears much improved.